# TabDPT-Turbo: Efficient In-Context Learning for Tabular Prediction

**Rasa Hosseinzadeh** [1]  **Alex Labach** [1]  **Zexin Xue** [1]  **Shuyi Han** [1]  **Valentin Thomas** [2]  **Anthony L. Caterini** [1]

## Abstract

Tabular foundation models, driven by in-context learning, have rapidly grown in quality and popularity. However, recent approaches with either cell-based architectures or retrieval have sacrificed efficiency for raw performance, restricting their utility in situations where compute is limited or inference speed is crucial. We adopt an alternate approach, sticking with row-based attention while incorporating long context pre-training to eliminate the need for retrieval. By combining this with architectural improvements and SSL pre-training on a newly-sourced, larger corpus of real data results, we present TabDPT-Turbo, a model that provides comparable default performance to TabDPT v1.1 on TabArena-Lite, CC18, and CTR23, at orders of magnitude faster. In our experiments, TabDPT-Turbo is the fastest model overall among leading foundation models. We have released the new model as TabDPT v1.2 at https://github.com/layer6ai-labs/TabDPT-inference.

## 1. Introduction

Tabular data remains the primary modality driving predictive AI in industry (van Breugel & van der Schaar, 2024). Tabular Foundation Models (TFMs), backed by in-context learning (ICL) have emerged in response, offering high-quality predictions without the need for extensive model training or hyperparameter tuning (Hollmann et al., 2023; Qu et al., 2025). However, many recent TFMs rely on expensive operations such as retrieval (Ma et al., 2025; Zhang et al., 2025b) or cell-based attention (Hollmann et al., 2025) to push performance at the cost of efficiency. This reduces their scope, limiting effectiveness in resource-constrained environments or low-latency settings.

This work instead revisits row-based TFMs (Hollmann et al., 2023). Efficiency and low resource deployment are typically easier to attain with row-based rather than cell-based architectures, particularly as context length increases. Besides the immediate benefits, enabling a more performant architecture with efficiency as a core design principle also provides the flexibility to perform inference-time adjustments such as fine-tuning and larger-scale explainability.

Considering these benefits, we directly extend TabDPT (Ma et al., 2025) as it is the highest-performing row-based architecture on the standard TabArena benchmark (Erickson et al., 2025) and is fully open-sourced. We pre-train on real data with the same Self-Supervised Learning (SSL) procedure, albeit with a corpus sourced from OpenML (Vanschoren et al., 2014) which is an order of magnitude larger than the previous pre-training set used in TabDPT. We also incorporate numerous architecture changes to improve model performance, including optimizations specifically developed for long-context training which enables us to avoid retrieval. While retrieval has been shown to improve TFM performance by providing a tailored local context to each test example (Thomas et al., 2024; Ma et al., 2024), it also greatly increases inference compute and memory requirements. Furthermore, the relevance of retrieval becomes less clear in the presence of long contexts; it has been shown in forthcoming work (Shaheen et al., 2026) that ICL-based TFMs learn nearest-neighbour-like behaviour internally. Our resulting model, TabDPT-Turbo, is an accelerated version of TabDPT with matching or better performance compared to previous versions. We summarize our contributions below:

- We present TabDPT-Turbo: a retrieval free, long context, row-based TFM aimed at efficient inference.
- We provide a new pre-training data corpus based on filtered and deduplicated OpenML tables, preserving the TabDPT-style SSL objective while scaling the amount and diversity of training data.
- We introduce numerous architecture and loss changes: attention temperature scaling, per-layer target routing through transformer value projections, learned thinking rows, regression-as-classification with Continuous Ranked Probability Score (CRPS), and an auxiliary context prediction loss.
- Results on TabArena-Lite (Erickson et al., 2025), CC18 (Bischl et al., 2021), and CTR23 (Fischer et al.,

---

[1]Layer 6 AI, Toronto, Canada  [2]Cohere, Toronto, Canada, work done while at Layer 6 AI. Correspondence to: <{rasa,alex,anthony}@layer6.ai>.

*Proceedings of the 2^{nd} ICML Workshop on Foundation Models for Structured Data*, Seoul, South Korea. 2026. Copyright 2026 by the author(s).

2023) show better predictive performance than Tab-DPT v1.1 at **orders of magnitude faster** inference, resulting in the fastest model among leading TFMs.

## 2. Related Work

**Tabular Foundation Models** TFMs are an active area of research, becoming dominant in tabular predictive benchmarks (Erickson et al., 2025), with TabPFN variants (Hollmann et al., 2023; 2025), TabICL and TabICLv2 (Qu et al., 2025; 2026), TabDPT (Ma et al., 2025), and many other strong models being introduced recently (Zhang et al., 2025a;b; Bouadi et al., 2025). But for a given compute budget, traditional or supervised tabular models can still provide competitive trade-offs, with Random Forest (Breiman, 2001), EBM (Lou et al., 2013), CatBoost (Prokhorenkova et al., 2018), and TabM (Gorishniy et al., 2025) all appearing on TabArena Pareto curves at the time of writing. A number of major design decisions affect TFM compute usage, and so we approach them with a focus on efficiency.

**Cell- vs. Row-Based Architectures** Many recent TFMs opt for cell-based attention mechanisms (Grinsztajn et al., 2025; Zhang et al., 2025b) instead of the classic row-based architecture popularized by TabPFNv1 (Hollmann et al., 2023). Indeed, a cell-based tokenization is expressive for heterogeneous columns and gets closer to encoding column invariance, which is often considered a desirable property for TFMs, but sequence length scales with $n \times m$ for $n$ rows and $m$ columns. We instead opt for a row-based setup as a deliberate efficiency choice: since each row is one token, long contexts become more feasible and inference remains easier to batch. Qu et al. (2026) use a hybrid approach, first using a lower-dimensional cell-based transformer that essentially acts as an encoder before passing through a row-based transformer; this is still slower than fully row-based models when the number of features increases.

**Retrieval** Several existing models, including TabDPT, use retrieval to select instances to include in the context (Thomas et al., 2024; Ma et al., 2025; Zhang et al., 2025b). This operation allows the model to utilize more training instances, but requires a separate context for each inference instance, meaning instances cannot be batched along with a shared context. Using long contexts instead maintains approximate locality in TFMs via attention (Shaheen et al., 2026) and so we elect to remove the retrieval for more efficient inference.

## 3. Method

### 3.1. Data

We adopt TabDPT's general training approach as a starting point since it is an open source row-based architecture. While TabDPT was originally trained on 112 datasets,

the authors argued for the potential of scaling data to improve model performance. We follow this direction by scaling to **1,445** datasets. We also deduplicate with respect to TabArena, which was not done in previous TabDPT versions, providing additional robustness to our results.

To extend the training data, we retrieved all active datasets on OpenML (Bischl et al., 2025) and applied filtering and deduplication stages to derive our final training set. First, we used the deduplication code provided in the original TabDPT training repository to exclude all datasets that were possible duplicates or derivations of datasets in CC18, CTR23, and TabArena. Empirically, we observed that datasets with too few columns could harm performance, and therefore filtered out datasets with fewer than 10 columns. We also removed excessively large datasets to speed up training, as we did not observe any significant change in final performance when doing so. Specifically, we removed datasets with over 200 columns or a 300 MB file size. We also applied column-level filters as part of pre-processing, removing categorical columns with a cardinality over 100, and entirely constant columns.

Finally, we added another duplicate detection step among groups of datasets that had equal row and column count. We reordered each candidate table to make comparisons invariant to row and column order. Rows and columns were permuted so that the largest element appeared in the top-left corner, after which the first row and first column were sorted. If two reordered tables were within a fixed tolerance, we kept only one of them. We compare our pre-training corpus with TabDPT v1.1 in Table 2 in the appendix.

Beyond scaling for performance, extending the training data ensures that we have a diverse range of datasets (up to 5M rows), enabling us to train the model with longer context.

**Self-Supervised Setup** Our SSL procedure follows Tab-DPT v1.1 (Ma et al., 2025). There, a table is first randomly sampled, then a task is constructed from this table. A column is selected, and if it satisfies some basic quality checks, it can be used as either a regression or classification target. Some post-processing is used to introduce more diversity, such as randomizing/merging classes or applying random functions and normalizing for regression. A random subset of the remaining columns is used as features. Finally, a random subset of instances is selected to be used as context, and another as queries; the TFM is trained to predict the targets for the queries given the query features, along with the context features and targets. In this work we use context lengths up to 32k rows.

### 3.2. Architecture

We use a row-based transformer architecture for efficiency. Each table row is padded with zeros to 128 dimensions and

then encoded with a linear layer. This follows the TabDPT design (with 128 instead of 100) and avoids the cost of cell-level modelling, whose sequence length scales with both rows and columns. The trade-off is that the encoder is not intrinsically invariant to column order. However, the row-based design is more efficient, stable during training, and robust against representation collapse. It enables ensembling by permuting feature columns. For datasets with more than 128 features, the model must reduce the feature dimension with methods like PCA or column subsampling.

The transformer backbone is a pre-norm variant using a 512-dimensional embedding, 32 transformer layers, 8 attention heads, and SwiGLU blocks. We also prepend 64 learned thinking rows to the sequence as in Hollmann et al. (2025). These rows are soft tokens and are not passed through the linear encoder; instead, they participate in attention, providing additional latent computation. Following Tab-DPT's attention design, rows attend only to the context and thinking rows, preventing query-to-query information flow.

Target conditioning is injected inside each transformer layer rather than being simply added to the row embedding as opposed to TabDPT. Each layer has a small target encoder that maps context targets through an MLP to embeddings. These embeddings are concatenated to the corresponding context row representations in the attention value stream, while queries and keys remain functions of the row representations. This makes target information available to the model through values while keeping context and query row representations similar to each other.

Queries and keys are additionally normalized per head before scaled dot-product attention. The attention branch includes a learned sigmoid gate, computed per token and attention head, which multiplicatively gates the attention output before the output projection. We also scale the attention temperature as the context length grows to mitigate dispersion (Veličković et al., 2025).

The prediction head is an MLP with one hidden layer that maps the final row representation to a joint output vector with 16 classification logits and 2048 regression logits. For classification, the first entries are interpreted as class logits and normalized with a softmax over the active classes. For regression, the remaining logits parameterize a categorical distribution over uniform bins spanning 10 standard deviations around the mean (calculated from context samples). The softmax over these bins gives a predictive distribution, and the point prediction is obtained as the expected bin centre. We depict the architecture in Figure 3 in the Appendix.

### 3.3. Objective

For classification tasks, we train with cross-entropy over up to 16 classes, matching the first 16 logits of the joint prediction head. When a dataset has fewer than 16 classes, only the active class logits are used. For datasets with more than 16 classes during inference, prediction can be extended using the same digit-by-digit strategy as TabDPT.

For regression, targets are standardized, and the model predicts a categorical distribution over 2048 bins spanning the interval $[-10, 10]$ in standardized target space (akin to Balazadeh Meresht et al. (2025) and Hollmann et al. (2025)). We optimize the CRPS loss between the predicted cumulative distribution and the target CDF induced by the observed value. This avoids choosing an arbitrary smoothing bandwidth required by cross-entropy on the same task. The CRPS loss is a proper scoring rule (Landsgesell & Knoll, 2026) which encourages calibrated predictive distributions, offering more information than TabDPT's point estimates. At inference time, the scalar prediction is obtained as the expectation of the bin centres under the predicted distribution.

During training, the regression loss is automatically rescaled by the ratio of the two base losses to put classification and regression on a similar scale without manual tuning. We additionally use a z-loss on the query logits to prevent logit explosion and a cross-entropy loss on context rows. The context prediction loss encourages the representations used for context and query rows to remain similar.

### 3.4. Inference Without Retrieval

The row-based transformer, long-context training, context-dependent attention scaling, and gated attention allow the model to use large contexts. We ran the model without retrieval on datasets with up to 100k context rows, using a single shared sequence containing all context rows followed by all query rows. This removes the need to build a kNN index and avoids passing a separate context for every query point through the model. This makes inference substantially more efficient than TabDPT v1.1.

## 4. Experiments

We evaluate TabDPT-Turbo on TabArena-Lite, CC18, and CTR23. Since our goal is to provide efficient inference, we report the default setting without using the tuned or ensembled configurations reported by TabArena. Our evaluation employs 8 forward passes per dataset, matching the evaluation protocol used for top performing foundation models such as TabICLv2, the TabPFN family, and previous Tab-DPT versions.

In Figure 4 of the Appendix, we see that, on TabArena-Lite, TabDPT-Turbo ranks fourth, trailing only the most recent foundation models while outperforming TabDPT v1.1, XG-Boost, TabICL, and TabPFNv2. While TabDPT-Turbo does not claim the top spot in raw prediction accuracy, it offers near-peak performance and improves substantially over

TabDPT v1.1 at a low inference cost. TabDPT-Turbo's performance is comparable to models that are computationally costlier or need dataset-specific training.

We further compare TabDPT-Turbo against TabDPT v1.1 on CC18 and CTR23 in Figure 1. For each model, we vary the number of inference passes and plot predictive performance against measured wall-clock inference time. Because previous versions of TabDPT construct a query-specific context, their inference cost grows with the number of query points and retrieved examples. In contrast, TabDPT-Turbo uses a single shared context followed by all query rows, allowing the query set to be evaluated in one efficient, batched forward pass.

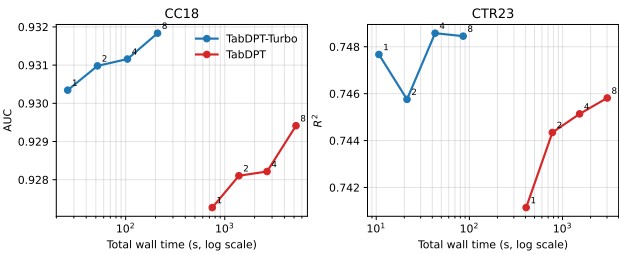

*Figure 1.* Performance versus total wall time for TabDPT-Turbo and TabDPT v1.1 on CC18 and CTR23 with $1, 2, 4$ and $8$ forward passes. $K = 2,048$ neighbours were used for TabDPT v1.1.

### 4.1. Inference Speed

Even with 8 forward passes per dataset, our total time for fitting and predicting is an average of 0.76s per 1000 instances. The existing TabArena-Lite results only report $k$-NN, ExtraTrees, and Random Forest methods as being faster, and notably report substantially slower total times for **all** gradient boosted tree ensembles or neural network models. While our hardware configurations differed, all models with comparable performance either require supervised training or use computationally heavier architectures, making it plausible for our method to be faster.

To evaluate our speed compared to other TFMs on the same hardware, we reran TabArena-Lite for TabDPT-Turbo as well as TabPFN-3 and TabICLv2, the fastest leading TFMs to the best of our knowledge. Results are shown in Table 1, with TabDPT-Turbo shown to be the fastest model.

*Table 1.* Average speeds on TabArena-Lite. All results generated with the same compute, using 1x H100 GPU and 96 vCPUs. All models use the default ensemble size of 8.

| Model | Fit time per 1k rows | Predict time per 1k rows |
| --- | --- | --- |
| TabDPT-Turbo | 0.44s | 0.19s |
| TabPFN-3 | 0.87s | 0.64s |
| TabICLv2 | 0.41s | 7.75s |

### 4.2. Ablations and Scaling

**Retrieval Ablation** Figure 2 compares retrieval at varying sample sizes $K$ against full-context inference on TabDPT-Turbo. On both classification and regression tasks, full context outperforms retrieval at every $K$ except $K = 8,192$, and even there the margin is small (mean AUC of 0.917 vs. 0.915, mean $R^2$ of 0.83 vs. 0.80). Meanwhile, retrieval is at least $100\times$ slower than full-context inference, and the only configuration that surpasses full context, $K = 8,192$, is nearly $1000\times$ slower. This suggests that the performance gap between full-context attention and nearest-neighbour retrieval is negligible, and that retrieval no longer justifies the inference-time cost it introduces.

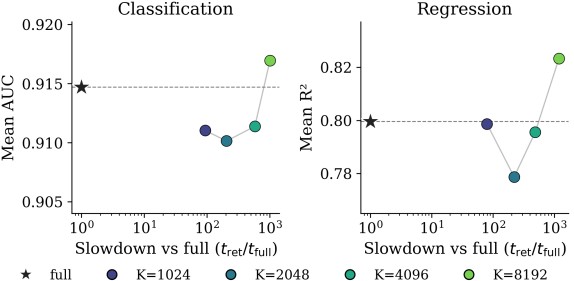

*Figure 2.* **Inference time vs. accuracy.** TabDPT-Turbo with no retrieval (star, dashed baseline) compared against retrieval with varying numbers of retrieved rows $K$ (circles). The x-axis shows inference slowdown relative to full-context mode ($t_{ret}/t_{full}$, log scale). Left: Mean AUC on CC18 and TabArena-Lite classification tasks. Right: Mean $R^2$ on CTR23 and TabArena-Lite regression tasks.

**Scaling** In Section A.4 of the Appendix, we investigate scaling both the model and context size, demonstrating that scaling both context length and parameter count leads to significant performance improvements.

## 5. Conclusion

While modern tabular ICL models have focused on maximizing predictive performance, this work pushes the boundaries of their efficiency. We present a row-based, retrieval-free, long-context model that enables computational trade-offs previously unavailable to neural network models.

Looking ahead, we plan to continue to improve the TabDPT model family in terms of predictive performance and speed, with this version providing a more effective starting point for scaling data and compute. We are also interested in leveraging it for understudied tabular data regimes, including datasets with large feature counts, large instance counts, and highly imbalanced data.

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

# A. Appendix

## A.1. Architecture Diagram

As described in the main text, the architecture uses transformer blocks as its backbone. Input features are first zero-padded to the required size and passed through a linear encoder. Thinking rows are then appended to the sequence as soft tokens. Targets are routed to the transformer value projections via per-layer MLPs rather than being added to the embedded rows. The output layer is split into classification and regression heads.

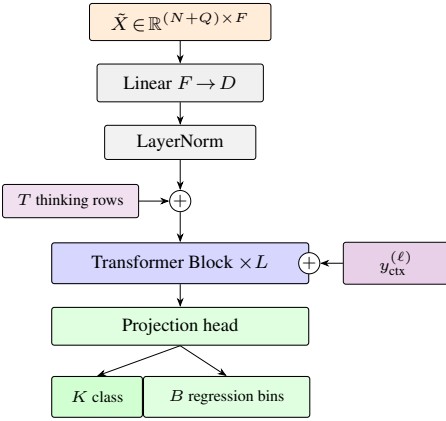

*Figure 3.* TabDPT architecture.

## A.2. Dataset comparison to TabDPT v1.1

*Table 2.* Corpus statistics: TabDPT v1.1 vs. TabDPT-Turbo

| Metric | TabDPT v1.1 | Ours | Ratio |
|---|---|---|---|
| Number of datasets | 112 | 1,445 | 12.90× |
| Total rows | 32.4M | 309.9M | 9.56× |
| Total features | 15.3K | 43.7K | 2.86× |
| Total cells | 0.87B | 5.60B | 6.41× |

## A.3. TabArena-Lite Elo score

Figure 4 provides the TabArena-Lite results for our models versus scores provided in the TabArena repository.

## A.4. Scaling

We investigate scaling along two dimensions: context size and model capacity. Throughout, $d$ denotes the hidden dimension, and $L$ denotes the number of layers.

**Context Scaling** We trained two models of identical architecture ($d = 256$, $L = 8$) but with different maximum context sizes: one limited to 1,024 input rows, the other extending to 16,384. As shown in Figure 5, the model trained with the larger context consistently outperforms its short-context counterpart, showing the benefit of increasing context size. Note that the total number of rows per batch was identical in these two settings.

**Model Scaling** We trained a medium-sized model with half the number of attention blocks as the full TabDPT-Turbo, giving both models a maximum context length of 32,768 for a fair comparison. As shown in Figure 6, TabDPT-Turbo (labelled "large") outperforms the medium model after roughly 500 epochs, indicating that scaling the parameter count yields meaningful gains.

We therefore see that scaling both context length and parameter count leads to significant performance improvements.

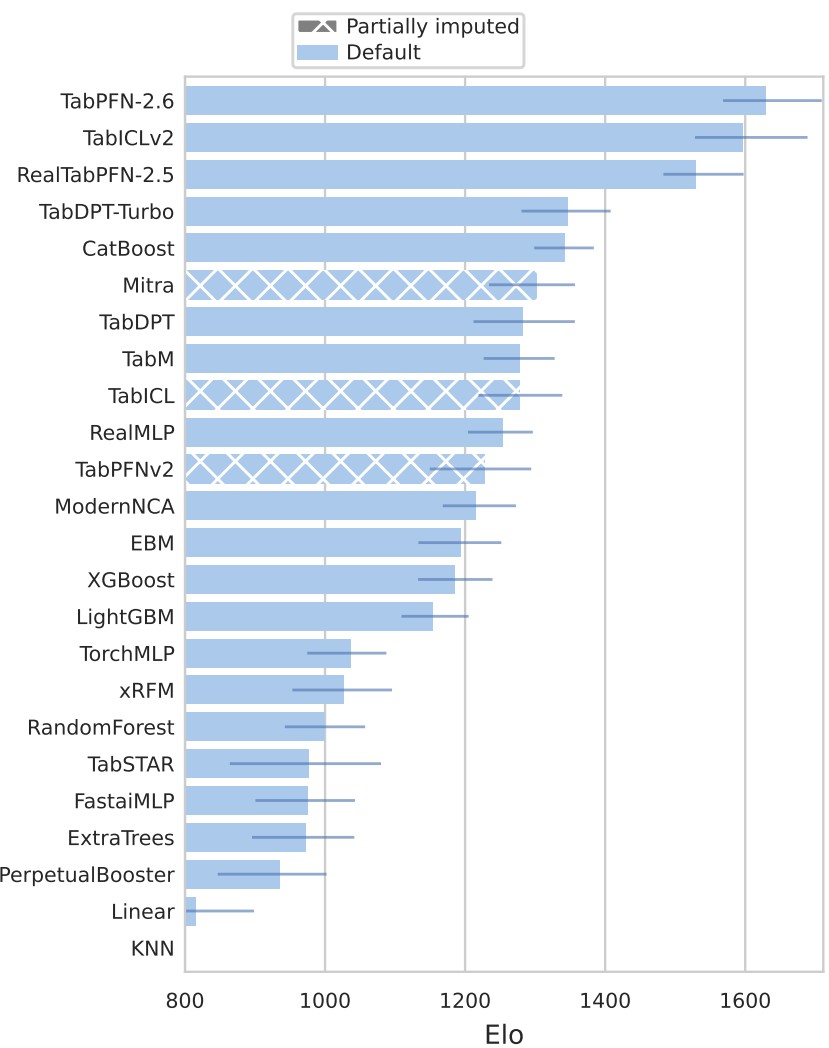

*Figure 4.* **TabArena-Lite Elo score comparison.** Evaluation is done with the default, non-tuned setting.

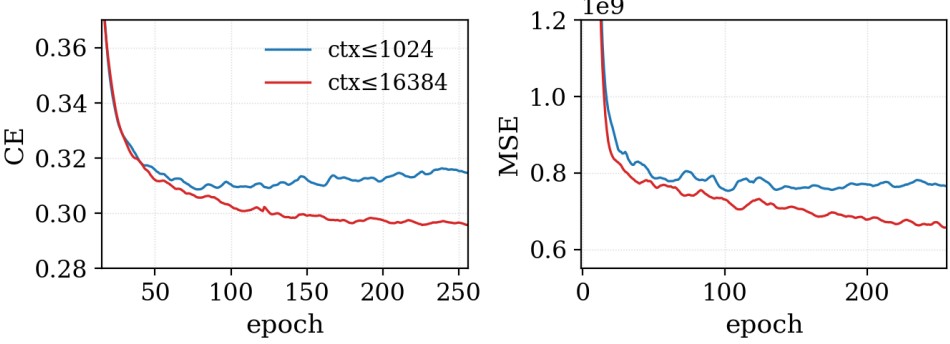

*Figure 5.* **Smaller vs. larger context.** Validation trajectories for two models with identical architecture but trained with different context sizes. Blue is trained on contexts of up to 1,024 rows and red up to 16,384.

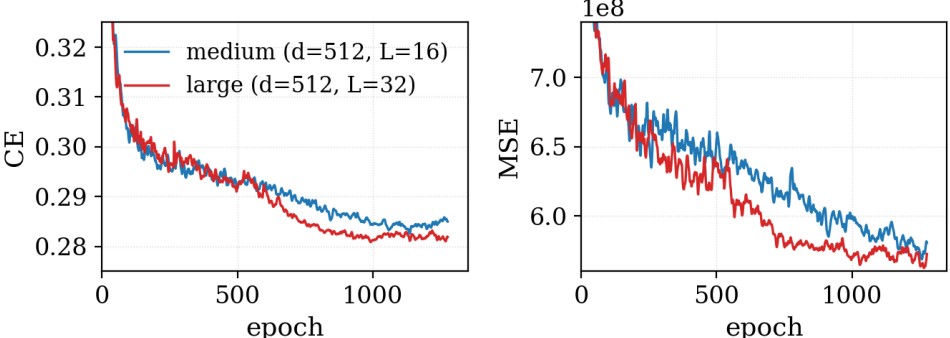

*Figure 6.* **Smaller vs. larger model.** Validation trajectories for the medium and large models. Left: cross-entropy loss on CC18 and TabArena-Lite classification tasks. Right: mean squared error on CTR23 and TabArena-Lite regression tasks.

