# OpenReview forum: "TabDPT-Turbo: Efficient In-Context Learning for Tabular Prediction"
_ICML.cc/2026/Workshop/FMSD — FMSD @ ICML 2026 Poster_

### Official Review · Reviewer_rUib · 2026-05-20
**TabDPT with more training data, better architecture, and no retrieval**

**Rating:** 8
**Confidence:** 5

**Review:**

This paper proposes a straightforward enhancement of TabDPT. The main changes are:
- Architecture improvements: 32 layers, SwiGLU + pre-norm, thinking tokens, target injection at each layer, gated attention, binned regression with CRPS loss instead of direct value prediction with L2 loss.
- Dropping context retrieval for faster inference.
- Training on more data (1445 tables from OpenML instead of 112 as in TabDPT).
- Training on longer context (up to 32k rows), and inference with up to 100k rows.

CC18 and CTR23 datasets are used mostly for validation, and TabArena for evaluation. Comparison is only run against default parameters, which penalizes models whose defaults are not optimized for this benchmark. In that setup, the results end up being worse than the three current leading models (TabPFN-v2.6, TabICL-v2, RealTabPFN-v2.5) but better than CatBoost and the original TabDPT.

The exposition is clear and rather straightforward. The multiple improvements over the original architecture suggest a significant amount of optimization work, which is beneficial for the community. There are no major breakthroughs, but the amount of innovation is more than enough for this workshop venue. As such, I think the paper should clearly be accepted.

I will still point out a few areas of improvement for the future.
- The paper extensively discusses a rather artificial "row-based" vs. "cell-based" distinction. Apparently, only models that first compress each row to a single token with an MLP end up being in the former class—therefore, essentially, TabPFN v1 and TabDPT. However, this is a restrictive framing. More modern models, like TabICL-v2, also reduce to a single token before applying transformers with their quadratic runtime. In fact, the asymptotics of the two approaches are exactly the same—the compression of cells into a single row token is O(num_cells) in both cases, and clearly that's a lower bound for any algorithm that at least reads the data. And for the end-to-end model, the asymptotics are O(nm + n^2) in both cases. As such, any difference between the two is merely a constant factor, which can be achieved either via different architecture or just with different hyperparameters. Therefore, the justification for using this specific architecture sounds post-hoc rather than a real motivation.
- It is unclear why large tables are excluded from training data. It would seem more reasonable to subsample them as aggressively as needed rather than throwing them away.
- While using 1.4k tables for training is certainly a larger scale, it sits at an uncomfortable middle ground between small scale (like TabDPT) and truly large scale (like, e.g., ConTextTab or Tabula-8B). At this scale, the contamination of test data into training is more dangerous: it's not few enough tables that they can be checked by hand, but not large enough (nor varied enough, since sourced from the same OpenML base as most benchmarks) that contamination does not matter. So one would need more details about how the contamination study was performed. With a scale of 1.4k training tables and 51 test tables, it would be feasible to just use an LLM to check them explicitly (for example, similar descriptions in OpenML or similar values). It's possible something like that was in fact done; if so, it is worth pointing that out in the paper (or the appendix).
- In a larger version of the paper, it would be interesting to have more ablation studies. Which of the many introduced changes were really significant for downstream results?
- One particularly interesting ablation concerns context length. The paper shows some ablation on training with 1k vs. 16k data points, but it only reports validation scores, not downstream task performance. The plots are also puzzling: it is not clear why training on shorter context would lead to more overfitting or worse validation scores, unless the training context size also affects validation. If so, that could be made clearer. In general, ablating and reporting results on a fixed downstream task is cleaner.
- Ideally, one would want a full 2D ablation of models trained on different context lengths and then tested with different context lengths on the downstream task. While multiple training runs are costly (but probably three or four would suffice, of which two are already present in the paper), evaluating on different context lengths is basically free.

---

### Official Review · Reviewer_yGBC · 2026-05-20
**Useful retrieval-vs-long-context evidence on a row-based TFM, weakened by bundled architectural changes**

**Rating:** 6
**Confidence:** 4

**Review:**

### *Summary*

TabDPT-Turbo extends TabDPT on two axes. First, a 12.9× larger OpenML-sourced pretraining corpus, with canonical-reordering deduplication against CC18, CTR23, and Tabarena. Second, a bundle of architectural changes on a row-based backbone trained with contexts up to 32k: attention temperature scaling, per-layer target routing through value projections, 64 learned thinking rows, gated attention, a CRPS regression head over 2,048 bins, an auxiliary context-prediction loss, and a z-loss on query logits.

The headline claim is that long-context training removes the need for retrieval at inference, so one batched forward pass serves all queries. On TabArena-Lite, CC18, and CTR23 (default setting, 8 forward passes), **TabDPT-Turbo matches or modestly beats TabDPT at lower inference cost.**

### *Strengths*

- Figure 3 (the retrieval ablation) is the strongest result in the paper. On a competitive row-based backbone, retrieval below K=8,192 loses to full-context inference, and even K=8,192 only marginally wins at roughly 1000× slowdown. This is the kind of clean head-to-head results that the space needs.
- The context-length ablation holds total rows-per-batch constant. That isolates the context-length effect from data throughput, which is the right control.
- The deduplication procedure (canonical reordering plus tolerance matching) goes beyond hash-based methods.
- CRPS sidesteps the bandwidth-selection problem of binned cross-entropy on regression. The automatic loss rescaling between classification and regression is a useful practical detail.
- Elo positioning is reported honestly: 4th on TabArena-Lite, behind TabPFN-2.6, TabICLv2, and RealTabPFN-2.5.

### *Areas for Improvement*

- Several architectural changes are introduced together with no **per-component ablation**. Attention temperature scaling, value-stream target routing, thinking rows, the CRPS head, gated attention, and the auxiliary context loss are each drawn from prior work, but it is currently unclear to me which components contribute most to the gains over TabDPT. As written, it is difficult to isolate what is driving the improvements beyond data scaling and retrieval removal.
- Pretraining cost is not stated. Given the 12.9× corpus expansion and 32k-context training setup, some estimate of total training compute would help practitioners evaluate the trade-off against the inference savings.
- The retrieval-vs-full-context conclusion depends on long-context training, but Figure 3 only runs on TabDPT-Turbo itself. I would also like to see the same retrieval ablation on the shorter-context (ctx=1024) variant, since that would help clarify whether the conclusion is specific to the long-context regime.
- Datasets with more than 128 features are handled by PCA or column subsampling. Alot of the benchmark tasks exceed this. The accuracy cost of the dimensionality reduction step is not reported.
- The "8 forward passes per dataset" protocol folds ensembling into the headline 0.76 s/1000-instances figure. Single-pass per-instance latency is closer to 0.1 s/1000. The split should be explicit, since most deployment budgets are per-call.
- TabM is not compared, which was a little surprising, since it appears prominently on TabArena Pareto curves and is an obvious non-foundation efficiency baseline.
- Figure 1 covers narrow accuracy ranges (CC18 AUC spread 0.004; CTR23 R² spread 0.006) with no error bars, so it is difficult to judge how significant the per-forward-pass gaps are.

### *Detailed Comments*

1. The attention temperature schedule from Velickovic et al. (2025) is not given explicitly. Reproducibility needs the formula, especially since context-length generalization is one of the paper's claims.
2. The per-layer target encoder MLPs in the value stream are a real departure from TabDPT's additive target embedding. Neither the architectural cost nor the contribution relative to a simpler additive baseline is measured.
3. The 2,048-bin regression head over [−10, 10] standardized space caps regression precision. State whether 2,048 was chosen empirically or by analogy to the paper by Balazadeh Meresht.
4. The z-loss on query logits and the auxiliary CE on context rows overlap with the per-layer target-routing motivation. I need clarification on whether they are mutually reinforcing or partly redundant.
5. Figure 1 uses K=2,048 for TabDPT. Reporting at K=8,192 (the setting that closes the accuracy gap in Figure 3) would give a fuller picture of where the wall-time advantage sits.
6. Whether retrieval becomes competitive again on very large training sets (e.g., 227k-row credit-card-fraud regimes) is left open and I believe that is worth flagging.
7. The deduplication procedure misses near-duplicates that differ by constant shift, scaling, or column-wise transformations. A brief false-negative discussion would strengthen the contamination claim.

### Justification of Score

Figure 3 is genuinely useful. For this backbone and evaluation setup, it provides strong evidence that retrieval below K=8,192 does not justify its cost, and the context-length and corpus-scaling results back up the broader claim that long-context training can replace retrieval at the scales evaluated. Scope sits well within in the workshop's agenda.

The architecture bundles known techniques without isolating which ones matter; the paper currently reads more as a successful combination of several ideas than as a study isolating which mechanisms are most important for efficient row-based TFMs. And the internal nearest-neighbour claim, which is the theoretical justification for dropping retrieval, rests on a forward reference no one can check. The retrieval result is compelling, though some of the broader architectural conclusions would benefit from additional disentangling experiments.

---

### Official Review · Reviewer_dbny · 2026-05-22
**Review: TabDPT-Turbo**

**Rating:** 7
**Confidence:** 3

**Review:**

**Summary.** The paper presents TabDPT-Turbo, an extension of TabDPT aimed at making row-based tabular foundation models substantially faster without sacrificing predictive performance. The core goal is efficiency: the authors remove retrieval, train with long contexts (up to 32k rows) so a single shared context can serve all queries in one batched forward pass, and scale the pre-training corpus roughly 10× by sourcing and filtering datasets from OpenML. They also introduce a handful of architectural and loss-side modifications. On TabArena-Lite, CC18, and CTR23, the resulting model matches or modestly exceeds TabDPT while running orders of magnitude faster at inference.

**Assessment.** This is an implementation paper rather than a research paper. Each of the individual ingredients is already established in the literature: long-context pre-training, learned thinking rows, attention temperature scaling, gated attention, SwiGLU blocks, regression-as-classification with a proper scoring rule like CRPS, and scaling up the pre-training corpus. The contribution lies in assembling these into a working, efficient pipeline. We nonetheless recommend acceptance: the engineering is solid, the efficiency gains are practically significant, and workshop venues are an appropriate home for this kind of consolidation work. The main weakness is the absence of ablations on the architectural and loss changes themselves. Modifications are introduced, but only context length and model size are ablated, so it is impossible to tell which changes actually drive the improvement over TabDPT. Adding per-component ablations would substantially strengthen the paper.

**Suggestion for turning this into a research paper.** The current work reads as a competent "look, I learned how to make a tabular foundation model" exercise, but nothing in it is genuinely new or striking. To move it into research territory, the authors could pick a specific observation or component and develop it into a real contribution. One concrete opportunity: the paper notes that "empirically, we observed that datasets with too few columns could harm performance, and therefore filtered out datasets with fewer than 10 columns." This is an interesting phenomenon that is currently sidestepped with a filter; investigating why it happens and proposing a method that lets the model train productively on low-column datasets would be a real research contribution. Alternatively, the authors could run thorough ablations of their architectural changes and provide a careful analysis of why each one helps or fails, turning the bag of tricks into a principled study of what actually matters in row-based tabular foundation models.